# Hematological and gene co-expression network analyses of high-risk beef cattle defines immunological mechanisms and biological complexes involved in bovine respiratory disease and weight gain

**Matthew A. Scott**[1]*, **Amelia R. Woolums**[2], **Cyprianna E. Swiderski**[3], **Abigail Finley**[1], **Andy D. Perkins**[4], **Bindu Nanduri**[5], **Brandi B. Karisch**[6]

**1** Veterinary Education, Research, and Outreach Center, Texas A&M University and West Texas A&M University, Canyon, TX, United States of America, **2** Department of Pathobiology and Population Medicine, College of Veterinary Medicine, Mississippi State University, Mississippi State, MS, United States of America, **3** School of Animal and Comparative Biomedical Sciences, University of Arizona, Tucson, Arizona, United States of America, **4** Department of Computer Science and Engineering, Mississippi State University, Mississippi State, MS, United States of America, **5** Department of Comparative Biomedical Sciences, College of Veterinary Medicine, Mississippi State University, Mississippi State, MS, United States of America, **6** Department of Animal and Dairy Sciences, Mississippi State University, Mississippi State, MS, United States of America

\* matthewscott@tamu.edu

**Data Availability Statement:** The data utilized in this study are found in the National Center for

## Abstract

Bovine respiratory disease (BRD), the leading disease complex in beef cattle production systems, remains highly elusive regarding diagnostics and disease prediction. Previous research has employed cellular and molecular techniques to describe hematological and gene expression variation that coincides with BRD development. Here, we utilized weighted gene co-expression network analysis (WGCNA) to leverage total gene expression patterns from cattle at arrival and generate hematological and clinical trait associations to describe mechanisms that may predict BRD development. Gene expression counts of previously published RNA-Seq data from 23 cattle (2017; n = 11 Healthy, n = 12 BRD) were used to construct gene co-expression modules and correlation patterns with complete blood count (CBC) and clinical datasets. Modules were further evaluated for cross-populational preservation of expression with RNA-Seq data from 24 cattle in an independent population (2019; n = 12 Healthy, n = 12 BRD). Genes within well-preserved modules were subject to functional enrichment analysis for significant Gene Ontology terms and pathways. Genes which possessed high module membership and association with BRD development, regardless of module preservation ("hub genes"), were utilized for protein-protein physical interaction network and clustering analyses. Five well-preserved modules of co-expressed genes were identified. One module ("steelblue"), involved in alpha-beta T-cell complexes and Th2-type immunity, possessed significant correlation with increased erythrocytes, platelets, and BRD development. One module ("purple"), involved in mitochondrial metabolism and rRNA maturation, possessed significant correlation with increased eosinophils, fecal egg count per

Biotechnology Information Gene Expression Omnibus (NCBI-GEO), accession number GSE161396. All data and code used for running experimental analyses are found on a GitHub repository at https://github.com/mscott16/2022-BRD-WGCNA and archived on Zenodo (DOI: 10.5072/zenodo.1015612). All remaining relevant data are found within the paper and its additional files.

**Funding:** This work was supported by the USDA National Institute of Food and Agriculture Animal Health and Disease Program (#2020-67016-31469). The funders had no role in study design, data collection and analysis, decision to publish, or preparation of the manuscript. The authors received no specific funding for this work.

**Competing interests:** The authors have declared that no competing interests exist.

gram, and weight gain over time. Fifty-two interacting hub genes, stratified into 11 clusters, may possess transient function involved in BRD development not previously described in literature. This study identifies co-expressed genes and coordinated mechanisms associated with BRD, which necessitates further investigation in BRD-prediction research.

## Introduction

Despite decades of research involved in discovering novel management tools, developing interventional systems, and advancing antimicrobial therapeutics, bovine respiratory disease (BRD) remains the leading cause of morbidity and mortality in beef cattle operations across North America [1–3]. Due to its widespread prevalence, BRD is considered one of the most economically devastating components of beef cattle production systems [2–4]. BRD is a polymicrobial, multifactorial disease complex, incorporating infectious agents, host immunity, and environmental elements as predisposing factors [5–7]. Previous research over the past several decades has greatly detailed these factors and risks associated with BRD, yet there is minimal evidence that overall rates of disease have improved [5, 8–10]. Furthermore, diagnostic evaluation of BRD often relies on visual signs attributed to the disease complex, which are commonly non-specific to airway and lung disease, and lack clinical sensitivity [11, 12]. Therefore, data driven approaches which capture the biological intricacies associated with clinical BRD development and provide candidate molecular targets capable of stratifying or predicting risk of disease and/or production loss would offer a more precise method of managing BRD.

Clinical BRD progression and severity often presents as an acute inflammatory disease [13]. However, molecular and cellular changes precede physiological changes in terms of disease development. As such, identifying consistent molecular and/or cellular components that relate to BRD development would allow for the development of rapid diagnostics capable of being performed with cattle at the time of facility arrival. Such a tool could facilitate precision medicine practices in stocker and feedlot operations and improve both speed and success of targeted therapy. Accordingly, hematological samples are ideal, as they represent a relatively noninvasive, cost effective, and readily obtainable source that reflects dynamic biological processes throughout the body [14, 15].

Previous research has investigated cellular and molecular components that may indicate or predict clinical BRD. Richeson and colleagues, utilizing complete blood count (CBC) variables and castration status at facility arrival, identified significant associations with BRD in calves with comparatively decreased numbers of eosinophils and increased numbers of erythrocytes [16]. When evaluating the relationships between cytokine gene expression and CBC data in cattle with concurrent BRD, Lindholm-Perry and colleagues discovered that cattle with BRD possessed a comparative increase in numbers of neutrophils, decrease in numbers of basophils, and increased expression of *CCL16*, *CXCR1*, and *CCR1* [17]. Recent RNA sequencing studies, performed by both our group and others, have identified mechanisms and candidate biomarkers in whole blood associated with BRD development [18–20]. However, these studies primarily sought to identify differentially expressed genes (DEGs) between cattle that were or were not treated for BRD based on clinical signs. Focus on identifying DEGs meant that much of the data generated by these studies was neglected. Therefore, we aimed to leverage global gene expression patterns across high-risk cattle, and incorporate available cellular-level hematological data from the same cattle, to infer mechanisms associated with BRD development with a more holistic approach.

As gene expression operates in tandem with biological regulatory networks and complexes, investigation of gene co-expression levels may reveal transcriptional coordination, distinguish protein production relationships, and measure cellular composition and function relevant to specific disease states such as BRD [21, 22]. This analysis approach falls into the field of systems biology, where, in contrast to reductionist biology, molecular components are pieced and scaled together to better understand disease and generate novel hypotheses [23, 24]. In this respect, we sought to build networks of co-expressed genes, utilizing the full structure of previously published gene expression data [20], and discover relationships between gene expression and cellular hematological components, which may elucidate and/or further confirm genes and mechanisms related to BRD development or resistance.

## Materials and methods

### Animal enrollment

All animal use and procedures were approved by the Mississippi State University Animal Care and Use Committee (IACUC protocol #17–120) and carried out in accordance with relevant IACUC and agency guidelines and regulations. This study was carried out in accordance with Animal Research: Reporting of In Vivo Experiments (ARRIVE) guidelines (https:// arriveguidelines.org). This study was conducted in accompaniment with previous work focused on differential gene expression analysis and candidate biomarker validation [20]; the RNA-Seq data of these animals were previously deposited in the National Center for Biotechnology Information (NCBI) Gene Expression Omnibus (GEO) database under accession number GSE161396. Briefly, 24 samples (n = 12 BRD, n = 12 Healthy) from the 2017 study were previously selected based on randomized stratification of vaccine and oral anthelminthic administration upon facility arrival (d0), and 24 samples from the 2019 study randomly selected with equal distribution of clinical BRD development within 28 days of arrival [20]. All cattle within each population (year) were of proportional arrival weight (S1 Table) and age (estimated 4–6 months). All animals enrolled in these two groups were commercial cattle, with unknown genetic characteristics and background; this is a typical attribute of newly received stocker cattle in commercial production systems. Of the 24 cattle from the 2017 population having RNA-Seq data, one individual (ID: 162–2017_S24; GSM4906455) was not incorporated into the network analysis due to missing CBC data. The following clinical data were recorded for each animal: at-arrival fecal egg counts per gram via modified-Wisconsin procedure (FEC-d0), body weight in pounds (WT) at arrival, Day 12, Day 26, and Day 82, average daily weight gain at each time point (ADG), growth rate (slope of weight over days recorded; GR), at-arrival castration status (Sex), at-arrival rectal temperature (Temp-d0), development of clinical BRD within 28 days post-arrival (BRD), number of clinical BRD treatments (Treat_Freq), and timing to first BRD treatment (Risk_Days). Ages (not recorded) were estimated to be similar upon facility arrival. Clinical data for these cattle are found in S1 Table.

### Hematology analysis

Approximately 6 mL of whole blood was collected at arrival into K$_3$-EDTA glass blood tubes (BD Vacutainer; Franklin Lakes, NJ, USA) via jugular venipuncture. Blood samples were stored at 4°C and analyzed the same day of collection with the flow cytometry-based Advia 2120i hematology analyzer (Siemens Healthcare Diagnostics Inc., Tarrytown, NY, USA), testing for the following parameters: white blood cells (WBC; K/μL), erythrocytes (RBC; M/μL), hemoglobin (HGB; g/dL), hematocrit (HCT; %), mean corpuscular volume (MCV; fL), mean corpuscular hemoglobin (MCH; pg), mean corpuscular hemoglobin concentration (MCHC; g/dL), red blood cell distribution width (RDW; %), and platelets (PLT; K/μL). Blood smear

staining was performed with a Hematek 3000 Slide Stainer (Siemens Healthcare Diagnostics Inc., Tarrytown, NY, USA) via Wright-Giemsa stain reagents. Stained blood smears were evaluated for leukocyte distribution via a manual 300-count white blood cell differential by trained clinical pathology technical staff at Mississippi State University College of Veterinary Medicine. Neutrophil, eosinophil, basophil, monocyte, and lymphocyte percentages were recorded, with accompanying neutrophil-to-lymphocyte ratios (NL Ratio). Hematology data for these cattle are found in S2 Table.

## RNA-Seq data processing and normalization

The gene-level raw count matrix generated from our previous research was utilized for this study [20]. Briefly, RNA was isolated via Tempus Spin RNA Isolation Kits (Thermo Fisher Scientific; Waltham, MA, USA), following manufacturer's protocol. TruSeq RNA Library Kit v2 (Illumina; San Diego, CA, USA) was utilized for mRNA sequencing library preparation, following manufacturer's protocol. Single-lane, high-throughput RNA sequencing was performed with NovaSeq 6000 S4 reagent kit and flow cell (Illumina). Sequence read files were quality assessed and trimmed with FastQC v0.11.9 [25] and Trimmomatic v0.39 [26], respectively. Reference-guided (*Bos taurus*; ARS-UCD1.2) read mapping, indexing, and gene-level assembly were performed with HISAT2 v2.2.1 [27, 28] and StringTie v2.1.2 [29, 30], respectively. The python program prepDE.py [31] was utilized for gene-level count matrix construction.

Raw gene counts were imported to R v4.0.4 and processed with the filterByExpr toolkit [32], removing genes with a minimum total count of less than 200 and counts-per-million (CPM) below 1.0 across a minimum of 12 libraries. Libraries were normalized with the trimmed mean of M-values method (TMM) [33, 34] and converted into log2-counts per million values (log2CPM). A total of 12,795 genes were identified after count processing and were utilized for weighted network analysis.

## Weighted gene co-expression network analysis (WGCNA)

Weighted network analysis was performed with the R package WGCNA v1.70.3 [35]. Clinical and hematology trait data were compiled and aligned to each respective sample library. To remove any outlier sample, canonical Euclidean distance-based network adjacency matrices were estimated and used to identify outliers based on standardized connectivity. Estimated adjacency matrices had network connectivity standardized with the provided equation, where the z-score normalized network connectivity ($Z.k_\mu$) for each sample is calculated from mean-center scaling of the raw network adjacencies (k) [36]:

$$Z.k_\mu = scale(k)_\mu = \frac{k_\mu - mean(k)}{\sqrt{var(k)}}.$$

Samples with a standardized connectivity < -5.0 were considered outliers and to be removed from further analysis; no samples were considered outliers in this study (S1 Fig). An adjacency matrix was constructed from the calculated signed Pearson coefficients between all genes across all samples. We utilized signed networks as they better capture gene expression trends (up- and down-regulation) and classify co-expressed gene modules which improve the ability to identify functional enrichment, when compared to unsigned networks [24, 35–37]. Soft thresholding was used to calculate the power parameter (β) required to exponentially raise the adjacency matrix, to reach a scale-free topology fitting index ($R^2$) of >80%; β = 8 was selected for this study. The relationship between each unit β and $R^2$ is seen in S2 Fig. Co-expression modules were constructed with the automatic, one-step blockwiseModules

function within the WGCNA R package, using the following parameters: power = 8, corType = "pearson," TOMType = "signed," networkType = "signed," maxBlockSize = 12795, minModuleSize = 30, mergeCutHeight = 0.25, and pamRespectsDendro = FALSE; all other parameters were set to default. Constructed co-expression modules were assigned a color by the WGCNA R package, with any gene not assembling into a specific module placed in the "grey" module. Module-trait associations were identified with Pearson correlation between module eigengene (ME; first principal component of the co-expression matrix [38] and clinical and hematology data). Modules were considered weakly or strongly correlated with each trait having a p-value $\leq$ 0.10 and $|R| \geq$ 0.3 or p-value $\leq$ 0.05 and $|R| \geq$ 0.4, respectively. Color scaling was performed with the Bioconductor package viridis v0.6.1 [39] to allow ease of visual interpretation for individuals with color blindness.

## Cross-population module preservation analysis

Based on our previous work, it can be inferred that host gene expression captured at facility arrival is variable across BRD severity cohorts [20, 40, 41]. Therefore, we assessed whether the at-arrival co-expression patterns and modules found in this study were well preserved across an RNA-Seq data set from an independent population of cattle. We investigated cross-populational module preservation across the whole blood transcriptomes of cattle previously assessed for differential gene expression (GSE161396; 2019 population (n = 24)) with the modulePreservation function found within the WGCNA R package. The gene-level raw count matrix from previous analysis [20] was utilized and processed, filtered, and normalized in identical procedures as the 2017 RNA-Seq data set (see RNA-Seq data processing and normalization section); a total of 12,803 genes were identified in the 2019 data set after count processing and normalization. Permutation testing ($n$ = 200 permutations) was conducted to assess the significance of module preservation across the 2017 and 2019 RNA-Seq data sets, utilizing the two composite statistical measurements Zsummary and medianRank scores [36, 42]. Briefly, the identified modules within the test network are randomly permuted $n$ times, where, for each permuted index, the mean and standard deviation is calculated for defining the corresponding Z statistic [42, 43]. Through the combination of additional preservation statistics (average of Zdensity and Zconnectivity), the calculated Zsummary statistic determines the level of mean connectivity among all genes within a module (i.e., network density) across the two data sets [24, 42]. Higher Zsummary values indicate a stronger level of module preservation between data sets but is dependent on the number of genes within the module (i.e., module size) [42]. To further evaluate preservation in a module size-independent manner, medianRank scores are calculated from the mean connectivity and density measurements observed from each module and assigned a rank score [42]. Lower medianRank values indicate a stronger level of module preservation between data sets. For this study, any module possessing Zsummary $\geq$ 10 and medianRank $\leq$ 5 was considered highly preserved.

## Functional enrichment analysis of preserved modules

WebGestalt 2019 [44] (WEB-based Gene SeT AnaLysis Toolkit; accessed September 13, 2021) was utilized for over-representation analysis to identify enriched Gene Ontology (GO) biological processes, cellular components, molecular functions, and pathways from genes found in each module considered well preserved. Pathway enrichment analysis was performed with the pathway database Reactome [45]. Human (*Homo sapiens*) gene orthologs and functional databases were utilized for GO term and pathway enrichment analyses. Over-representation analysis parameters within WebGestalt 2019 included between 3 and 3000 genes per category, Benjamini-Hochberg (BH) procedure for multiple hypothesis correction, adjusted p-value

(FDR) cutoff of 0.05 for significance, and a total of 10 expected reduced sets of the weighted set cover algorithm for redundancy reduction.

## BRD-associated hub gene identification and network analyses

Hub genes are those genes found within a module (eigengenes) that possess high connectivity which may exhibit a greater degree of biological significance in respect to significantly associated clinical traits, when compared to all other eigengenes [38, 46, 47]. Here, we sought to identify hub genes found from modules which are significantly associated with any of the clinical BRD categories (BRD, Treat_Freq, and Risk_Days). This was performed in the WGCNA R package with two procedures. First, Pearson correlation between gene expression and module eigengenes was calculated, resulting in the level of module membership ($k_{ME}$) for each gene. Second, the Pearson correlation between individual gene expression level and clinical trait was calculated, resulting in the level of gene significance (GS) for each gene. Any gene possessing $k_{ME}$ and GS values $\geq 0.7$ and $\geq 0.3$, respectively, were considered hub genes for clinical traits [36]. All BRD-associated hub genes were used for network construction of known and predicted protein-protein interactions with the Search Tool for the Retrieval of Interacting Genes (STRING) database v11.5 [48], utilizing bovine (*Bos taurus*) annotations. STRING analysis was performed with the physical subnetwork setting, where edges only display protein interactions that have evidence of binding to or forming a physical complex. Any interaction above a combined score (confidence) of 0.200 was incorporated into the complete network prior to network clustering; disconnected nodes were removed from the network. The Markov Cluster (MCL) algorithm was utilized for network clustering due to its superior performance in complex extraction without the need of additional parameter tuning [49]. Hub genes within the interaction network were placed into distinct clusters based on MCL clustering of the distance matrix acquired from the combined interaction scores, using a MCL inflation parameter of 1.4.

## Statistical analysis

Clinical and hematology data (described in animal enrollment and hematology analysis) were compared between cattle treated for naturally-acquired clinical BRD within the first 28 days following facility arrival (BRD) and those never being diagnosed nor treated (Healthy). Residual normality was assessed in R v4.0.4 with the Shapiro-Wilk test [50], with an *a priori* level of significance set at 0.10; neutrophil percentage (Neu%), eosinophil percentage (Eos%), basophil percentage (Baso%), lymphocyte percentage (Lymph%), neutrophil-to-lymphocyte ratio (NL ratio), FEC-d0, MCHC, RDW, and Sex were considered non-normally distributed. Differences in normally distributed variables between BRD and Healthy cattle were assessed with the Student's *t*-test. Differences in non-normally distributed variables were assessed with the Welch's *t*-test; differences between the two groups with respect to Sex was assessed with Pearson's chi-square test with Yates' continuity correction. Differences between BRD and Healthy cattle were considered significant having a p-value $\leq 0.05$.

## Results

### Statistical analysis of clinical and hematological parameters

Descriptive statistics for the clinical and hematological data are provided in Table 1. Regarding the hematological parameters, average values of Lymph%, RDW, and PLT were outside of the internal reference intervals for both BRD and Healthy cattle. In this study, RBC was considered significantly higher at arrival in BRD cattle compared to Healthy cattle; no other parameter

**Table 1. Statistical analysis of hematological and clinical traits between BRD and healthy groups.**

| Variable | Internal Reference | BRD mean (s.d.) | Healthy mean (s.d.) | p-value |
|---|---|---|---|---|
| Neu% | 37.000–80.000 | 35.917 (5.547) | 37.213 (9.748) | 0.717 |
| Eos% | 0.000–12.000 | 3.944 (3.237) | 2.635 (1.616) | 0.251 |
| Baso% | 0.000–2.500 | 0.193 (0.213) | 0.151 (0.218) | 0.658 |
| Mono% | 0.000–12.000 | 8.862 (4.603) | 8.363 (4.507) | 0.805 |
| Lymph% | 10.000–50.000 | 51.083 (4.756) | 51.635 (11.928) | 0.893 |
| NL Ratio | N/A | 0.711 (0.141) | 0.859 (0.660) | 0.504 |
| WBC (K/µL) | 4.000–12.000 | 7.430 (2.722) | 7.320 (1.292) | 0.913 |
| RBC (M/µL) | 5.000–9.990 | 9.605 (0.568) | 9.032 (0.676) | 0.047 |
| HGB (g/dL) | 7.700–15.000 | 13.075 (1.071) | 12.491 (0.906) | 0.194 |
| HCT (%) | 25.000–45.000 | 36.125 (3.269) | 35.000 (2.534) | 0.391 |
| MCV (fL) | 36.000–55.000 | 37.725 (3.843) | 38.845 (2.851) | 0.460 |
| MCH (pg) | 12.000–22.000 | 13.625 (1.112) | 13.855 (0.806) | 0.597 |
| MCHC (g/dL) | 32.000–40.000 | 36.225 (1.190) | 35.691 (0.977) | 0.272 |
| RDW (%) | 11.600–14.800 | 29.258 (2.362) | 27.564 (3.023) | 0.171 |
| PLT (K/µL) | 200.000–900.000 | 1413.083 (506.885) | 1149.000 (401.516) | 0.203 |
| FEC-d0 | N/A | 761.250 (768.795) | 618.364 (408.492) | 0.597 |
| ADG-d12 | N/A | 0.667 (1.604) | 2.167 (1.838) | 0.059 |
| ADG-d26 | N/A | 1.917 (1.204) | 2.710 (0.948) | 0.110 |
| ADG-d82 | N/A | 2.273 (0.599) | 2.946 (0.432) | 0.008 |
| Growth Rate | N/A | 2.370 (0.554) | 2.995 (0.435) | 0.009 |
| Temp-d0 (F˚) | N/A | 103.333 (0.712) | 103.291 (0.667) | 0.890 |
| Sex | N/A | 10 bulls, 2 steers | 10 bulls, 1 steer | 1.000 |

Means, standard deviations (in parentheses), and statistical probability values of differences in hematological and clinical parameters between BRD (n = 12) and Healthy (n = 11) cattle. Parameters were considered significantly different with p-values ≤ 0.05.

was considered significantly different between the two groups. Regarding clinical data, BRD cattle possessed significantly lower weight gain by end of study (ADG-d82; 2.273 lbs/day in BRD and 2.946 lbs/day in Healthy) and lower calculated slopes of weight gain over time (Growth Rate; 2.370 in BRD and 2.995 in Healthy); no other clinical parameter was considered significantly different between the two groups. We did not include the at arrival weight (WT-d0) as an analysis variable because there was no significant difference between the Healthy (mean: 477.0, s.d.:24.8) and BRD (mean: 475.3, s.d.:26.9) cohorts.

## Weighted gene co-expression network construction

The remaining filtered genes (n = 12,795) were used for WGCNA network and module construction. The resulting network identified a total of 41 color-coded modules of co-expressed genes, excluding the grey module which incorporates uncorrelated genes (n = 1,235) (Fig 1). Across the 41 assigned modules, the turquoise module possessed the largest number of co-expressed genes (n = 2,503) and the lightsteelblue1 module possessed the smallest number of co-expressed genes (n = 38); the average size of each module was approximately 282 genes. The complete list of genes and module assignment is found in S3 Table.

Automated block-wise module detection of interconnected genes were grouped into 41 unique color-coded modules, excluding the grey module (uncorrelated genes). The x-axis corresponds to the gene-module assignment and the y-axis (Height) depicts the calculated distance between co-expressed genes from hierarchical average linkage clustering.

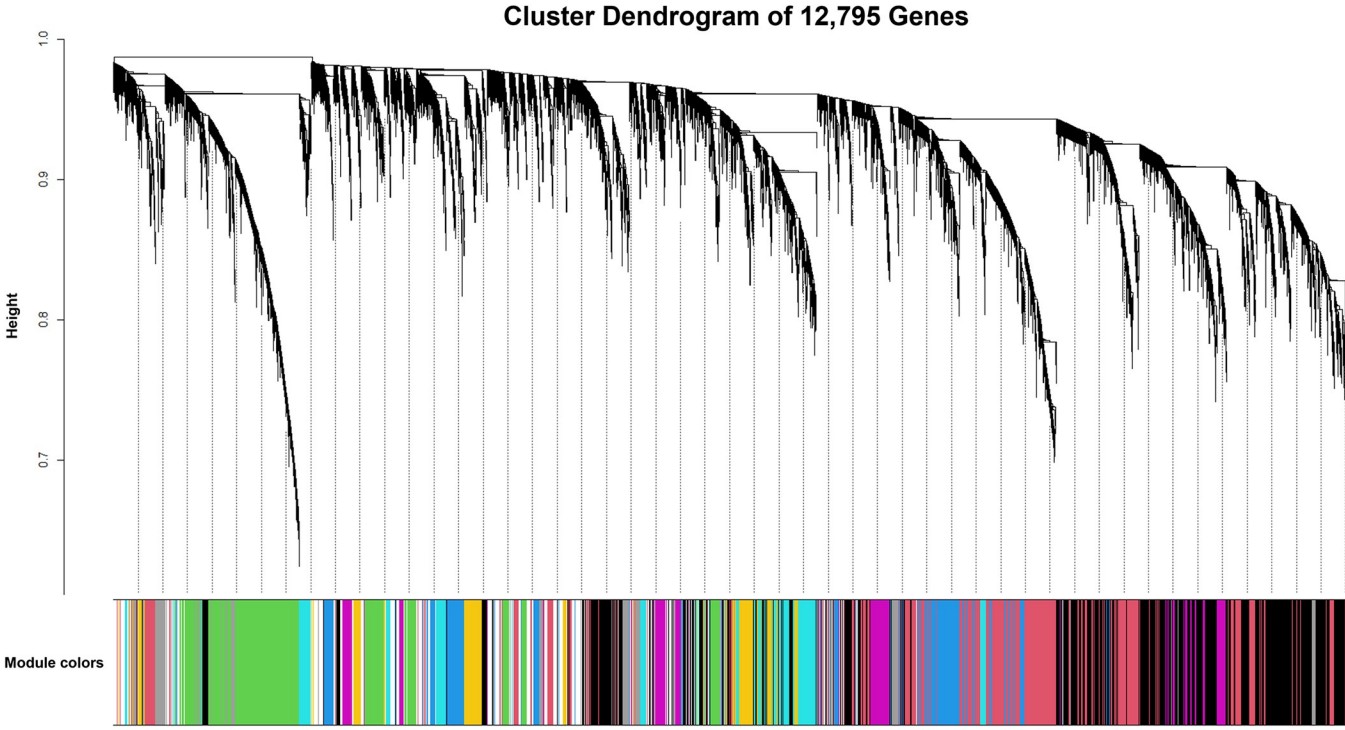

**Fig 1. Cluster dendrogram of 12,795 genes generated through dissimilarity metrics (1-TOM) and hierarchical clustering.**

## Module-trait relationship with hematological and clinical datasets

Pearson correlation heatmaps were generated to assess the relationship between all modules and hematological clinical datasets. Regarding hematological data, several significant relationships of interest exist (Fig 2). The tan module possessed the highest number of significant correlations with the hematological data (8), followed by turquoise, pink, lightgreen, and lightcyan modules (7). With respect to RBC, considered significantly higher at arrival in BRD cattle compared to Healthy cattle, six modules were strongly correlated: paleturquoise ($R = 0.44$, $p = 0.03$), lightcyan ($R = 0.51$, $p = 0.01$), green ($R = 0.41$, $p = 0.05$), steelblue ($R = 0.50$, $p = 0.01$), brown ($R = -0.45$, $p = 0.03$), turquoise ($R = 0.49$, $p = 0.02$). Additionally, seven modules were considered weakly correlated with RBC: magenta ($R = 0.32$, $p = 0.10$), darkgreen ($R = 0.36$, $p = 0.09$), lightsteelblue1 ($R = -0.36$, $p = 0.09$), blue ($R = 0.36$, $p = 0.10$), saddlebrown ($R = 0.36$, $p = 0.09$), orangered4 ($R = -0.33$, $p = 0.10$), tan ($R = -0.36$, $p = 0.09$). Regarding modules correlating with RBC, three modules possessed significant associations with multiple related red cell indices (HGB, HCT, MCV, MCH, MCHC, and RDW): saddlebrown, steelblue, and lightcyan. Saddlebrown was strongly associated with MCV ($R = -0.63$, $p = 0.001$) and MCH ($R = -0.62$, $p = 0.001$), and weakly associated with HCT ($R = -0.31$, $p = 0.10$) and MCHC ($R = 0.36$, $p = 0.10$). Steelblue was strongly associated with RDW ($R = 0.70$, $p = 2e-04$) and weakly associated with HGB ($R = 0.35$, $p = 0.10$) and MCHC ($R = 0.40$, $p = 0.06$). Lightcyan was strongly associated with HGB ($R = 0.47$, $p = 0.02$) and RDW ($R = 0.51$, $p = 0.01$) and weakly associated with HCT ($R = 0.38$, $p = 0.08$).

Pearson correlations between each of the unique color-coordinated modules and hematological traits are visualized and represented as a heatmap. Each row represents a distinct co-expression module, and each column represents hematological traits as follows: white blood cells (WBC; K/μL), erythrocytes (RBC; M/μL), hemoglobin (HGB; g/dL), hematocrit (HCT;

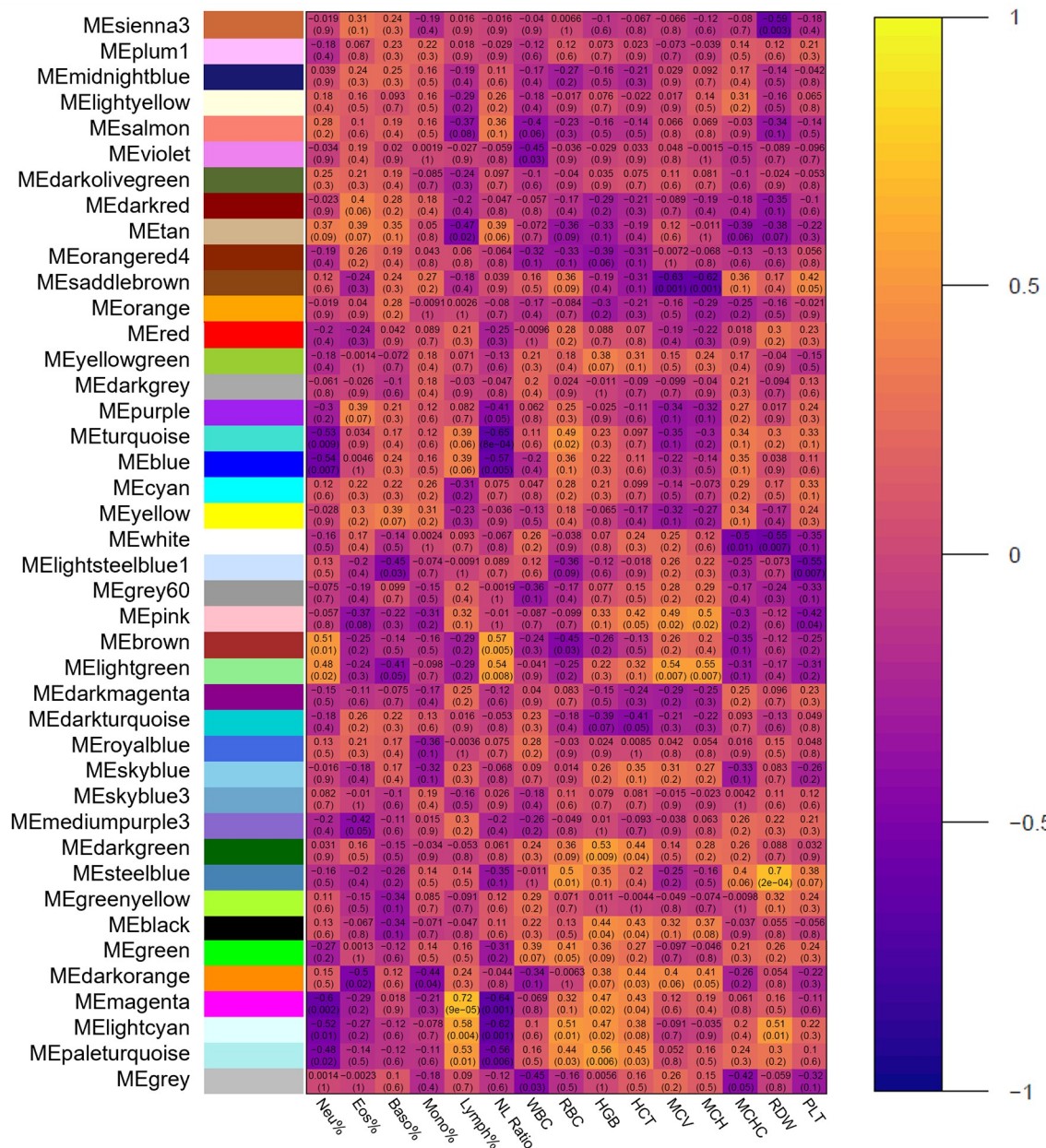

**Fig 2. Module-trait relationships between co-expression modules and hematological traits.**

%), mean corpuscular volume (MCV; fL), mean corpuscular hemoglobin (MCH; pg), mean corpuscular hemoglobin concentration (MCHC; g/dL), red blood cell distribution width (RDW; %), and platelets (PLT; K/μL). Cells are represented by how positive (yellow/white) or negative (purple/black) the correlation is between module and hematological trait, respectively.

The relationships between modules and clinical data are found in Fig 3. With respect to all clinical disease associations (BRD, Treat_Freq, and Risk_Days), five modules possessed significant correlations: steelblue, mediumpurple3, royalblue, orange, and violet. Steelblue was strongly associated with BRD ($R$ = 0.41, $p$ = 0.05) and Risk_Days ($R$ = -0.41, $p$ = 0.05).

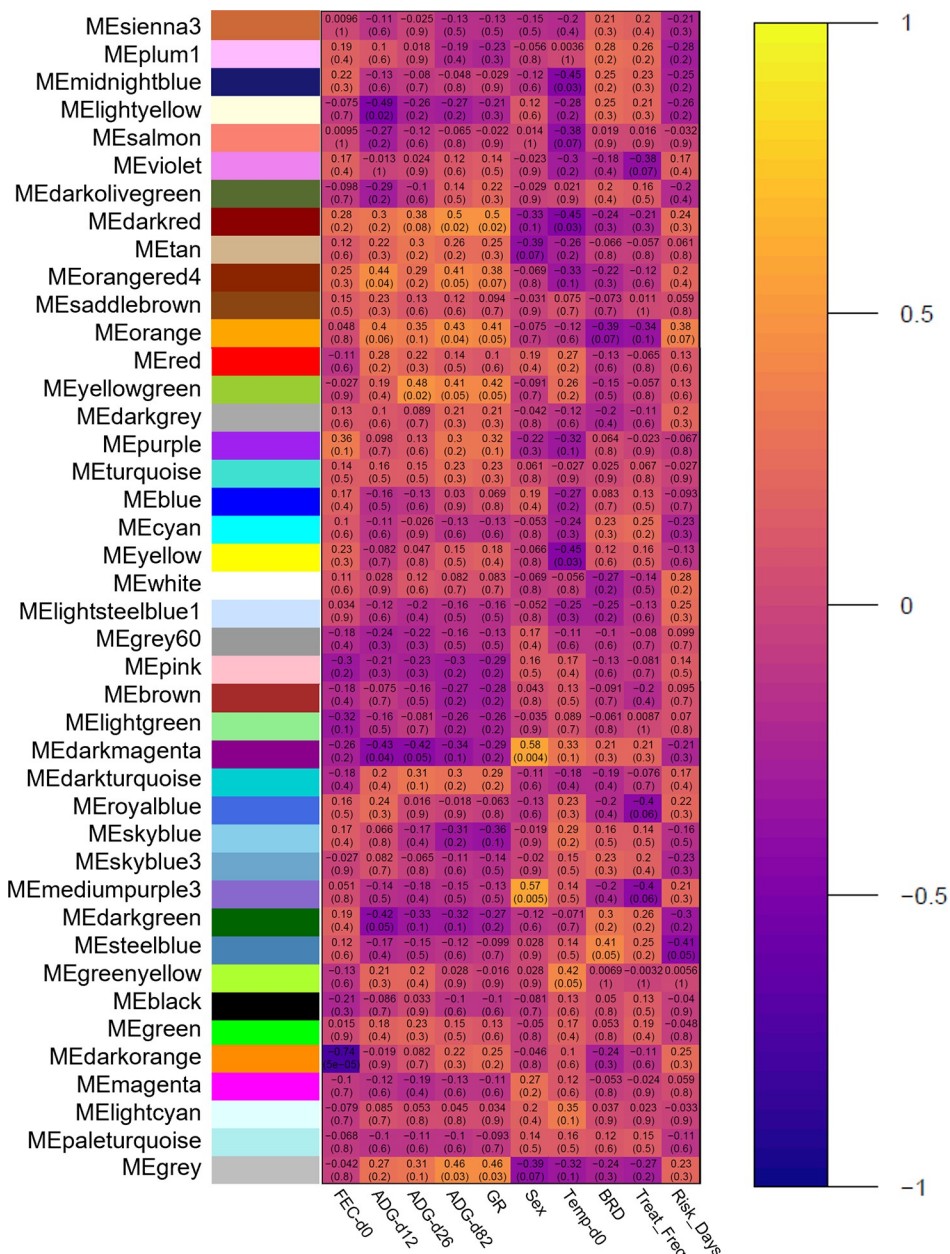

**Fig 3. Module-trait relationships between co-expression modules and clinical traits.**

Mediumpurple3 was weakly associated with Treat_Freq ($R$ = -0.40, $p$ = 0.06). Royalblue was weakly associated with Treat_Freq ($R$ = -0.40, $p$ = 0.06). Orange was weakly associated with BRD ($R$ = -0.39, $p$ = 0.07), Treat_Freq ($R$ = -0.34, $p$ = 0.10), and Risk_Days ($R$ = 0.38, $p$ = 0.07). Violet was weakly associated with Treat_Freq ($R$ = -0.38, $p$ = 0.07). Regarding production traits (ADG-d12, ADG-d26, ADG-d82, and GR), ten modules possessed significant correlations: darkgreen, skyblue, darkturquoise, darkmagenta, purple, yellowgreen, orange, orangered4, darkred, and lightyellow. However, to mitigate unexplained variation which may confound differences in ADG-d12 and ADG-d26, coupled with the lack of significance between disease cohorts, eight modules correlating with ADG-d82 and GR were prioritized. Darkred was

strongly associated with ADG-d82 ($R = 0.50$, $p = 0.02$) and GR ($R = 0.50$, $p = 0.02$). Orangered4 was strongly associated with ADG-d82 ($R = 0.41$, $p = 0.05$) and weakly associated with GR ($R = 0.38$, $p = 0.07$). Orange was strongly associated with ADG-d82 ($R = 0.43$, $p = 0.04$) and GR ($R = 0,41$, $p = 0.05$). Yellowgreen was strongly associated with ADG-d82 ($R = 0.41$, $p = 0.05$) and GR ($R = 0.42$, $p = 0.05$). Purple was weakly associated with GR ($R = 0.32$, $p = 0.10$). Darkmagenta was weakly associated with ADG-d82 ($R = -0.34$, $p = 0.10$). Skyblue was weakly associated with GR ($R = -0.36$, $p = 0.10$). Darkgreen was weakly associated with ADG-d82 ($R = -0.32$, $p = 0.10$). Notably, orange was the only module which possessed significant correlations with both disease-associated and weight gain traits. However, orange did not possess any significant correlations with hematological traits.

Pearson correlations between each of the unique color-coordinated modules and clinical traits are visualized and represented as a heatmap. Each row represents a distinct co-expression module, and each column represents clinical traits as follows: at-arrival fecal egg counts per gram via modified-Wisconsin procedure (FEC-d0), body weight in pounds (WT) at arrival, Day 12, Day 26, and Day 82, calculated average daily weight gain at each time point (ADG), growth rate (slope of weight over days recorded; GR), at-arrival castration status (Sex), at-arrival rectal temperature (Temp-d0), development of clinical BRD within 28 days post-arrival (BRD), number of clinical BRD treatments (Treat_Freq), and timing to first BRD treatment (Risk_Days). Cells are represented by how positive (yellow/white) or negative (purple/black) the correlation is between module and clinical trait, respectively.

## Cross-populational network preservation analysis

Module preservation analysis identified five modules considered well preserved across the 2017 and 2019 populations: black (size = 432; Zsummary = 39.772; medianRank = 4), purple (size = 296; Zsummary = 34.773; medianRank = 2), lightgreen (size = 123; Zsummary = 23.291; medianRank = 1), tan (size = 222; Zsummary = 17.559; medianRank = 5), and steelblue (size = 59; Zsummary = 11.555; medianRank = 3) (Fig 4). Notably, steelblue was the only well-preserved module which possessed significant association with BRD-related clinical traits.

The medianRank and Zsummary values across all modules are depicted through the scatterplot x- and y-axes, respectively. Zsummary values $\geq 10.0$ and medianRank values $\leq 5.0$, indicated by dashed lines, denote that a module identified with the 2017 gene expression data is well preserved across the 2019 gene expression data.

## Functional enrichment analysis of well-preserved modules

To explore the functionality and biological relevance of the five well preserved modules, we performed over-representation analysis with all genes from each module (black, purple, lightgreen, tan, and steelblue; S4 Table). Analysis of genes from the black module revealed 47 biological process terms, 49 cellular component terms, 17 molecular function terms, and five significantly enriched pathways. Biological processes identified from genes within the black module were related to neutrophil activity and degranulation, aldehyde metabolism, nitrogen compound response and catabolism, and cellular transport. Cellular components identified from genes within the black module involved intracellular and extracellular vesicles, secretory granules, cellular junctions, and lysosomes. Molecular functions identified from genes within the black module involve cytokine, enzyme, and calcium-dependent protein binding, aldehyde dehydrogenase (NAD) activity, and interleukin-1 receptor activity. Enriched pathways identified from genes within the black module involved neutrophil degranulation, metabolic disease, and signaling via tyrosine kinase receptor.

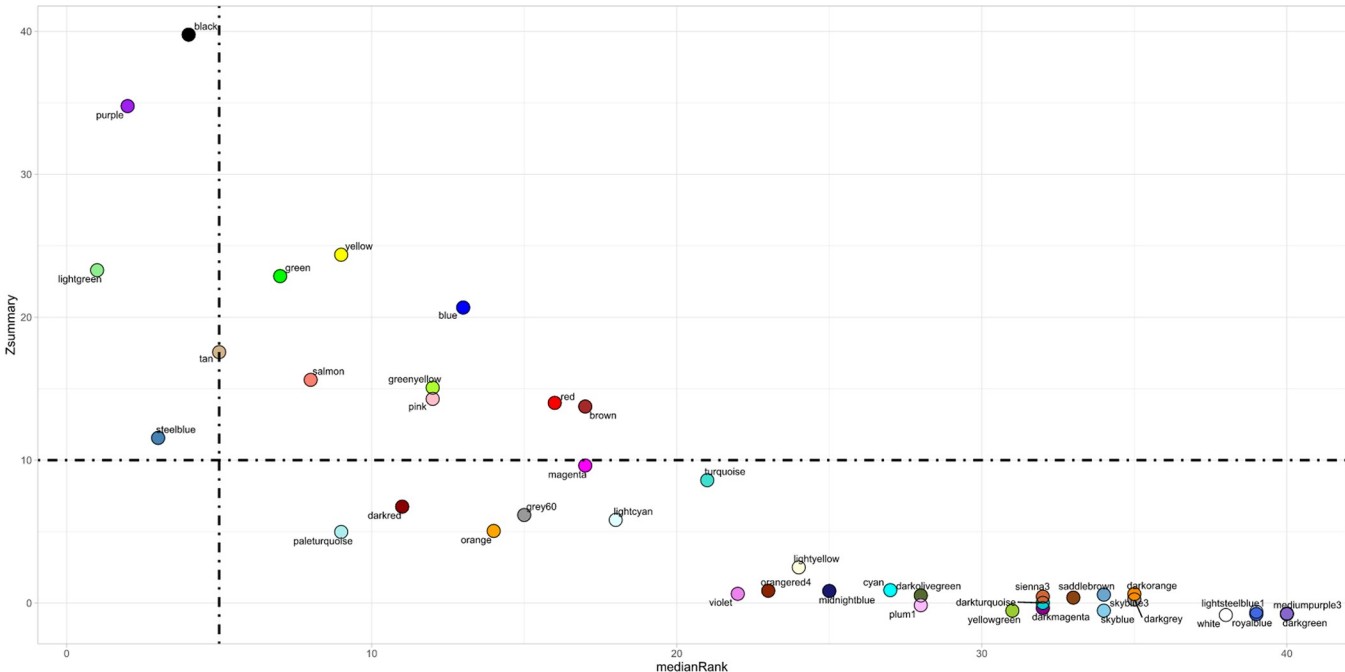

**Fig 4. Cross-populational module preservation analysis.**

Analysis of genes from the purple module revealed 54 biological process terms, 46 cellular component terms, 16 molecular function terms, and 40 significantly enriched pathways. Biological processes identified from genes within the purple module involved mitochondrial processes (cristae formation, respiratory chain complex assembly), non-coding RNA processing and maturation, cellular protein transport, and metabolic processes and biosynthesis. Cellular components identified from genes within the purple module involved cell substrate and adhesion junction, ribosomes, cytoplasmic side of endoplasmic reticulum, mitochondrial inner membrane and envelope, and the 48S preinitiation complex. Molecular functions identified from genes within the purple module involved mRNA/rRNA binding, ubiquitin ligase inhibition, ATP synthase activity, and NADH dehydrogenase. Enriched pathways identified from genes within the purple module involved infectious disease/viral infection, amino acid metabolism, translation initiation/termination, rRNA processing, and ATP synthesis and respiratory electron transport.

Analysis of genes from the lightgreen module revealed 38 biological process terms, 49 cellular component terms, three molecular function terms, and one significantly enriched pathway. Biological processes identified from genes within the lightgreen module involved leukocyte/neutrophil differentiation, activation, and degranulation, tissue remodeling, cell secretion and exocytosis, phagocytosis and micropinocytosis, dendritic cell activation, and interleukin-8 secretion. Cellular components identified from genes within the lightgreen module involved lysosome, secretory/azurophil granule, vesicular/vacuolar membrane, granule lumen, and macropinosome. Molecular functions identified from genes within the lightgreen module involved symporter activity, potassium-chloride symporter activity, and phosphatidylinositol binding. The single enriched pathway identified from genes within the lightgreen module was neutrophil degranulation.

Analysis of genes from the tan module revealed 35 biological process terms, 32 cellular component terms, four molecular function terms, and two significantly enriched pathways.

Biological processes identified from genes within the tan module involved B-cell activation, receptor signaling, and regulation, immunoglobulin production, cytokine production, positive regulation of interferon-gamma production, and mononuclear cell proliferation. Cellular components identified from genes within the tan module involved MHC class II protein complex, lytic vacuole membrane, clathrin-coated endocytic vesicle, endosomal membrane, and B-cell receptor complex. Molecular functions identified from genes within the tan module involved MHC class II receptor activity, MHC class II protein complex binding, and peptide antigen binding. Enriched pathways identified from genes within the tan module were antigen activates B-cell receptor (BCR) leading to generation of second messengers and CD22-mediated BCR regulation.

Analysis of genes from the steelblue module revealed three biological process terms, three cellular component terms, no molecular function terms, and no significantly enriched pathways. Biological processes identified from genes within the steelblue module were cell surface receptor signaling pathway, negative regulation of fibroblast growth factor receptor signaling pathway, and antigen receptor-mediated signaling pathway. Cellular components identified from genes within the steelblue module involved side of membrane, plasma membrane part, and alpha-beta T cell receptor complex.

## BRD-associated hub gene identification and *in silico* protein-protein interaction and clustering analyses

Hub gene identification analysis included co-expressed genes from the following modules: violet (54), orange (68), royalblue (100), mediumpurple3 (41), and steelblue (59). The $k_{ME}$ and GS value cutoffs within each module resulted in 24, 46, 30, 22, and 32 BRD-associated hub genes from the violet, orange, royalblue, mediumpurple3, and steelblue modules, respectively (S5 Table). These resulting hub genes were further utilized for physical subnetwork protein-protein interactions and network clustering. After removal of all disconnected nodes, the interaction network demonstrated significant connectivity between 52 proteins across 11 distinct clusters with high inter-nodal connectivity (Fig 5); these gene products and their combined interaction scores are found in S6 Table. These connected gene products demonstrate possible at-arrival biomolecular complexes associated with BRD development and severity.

Interaction score analysis reveals 52 genes, with high intramodular and BRD-trait relationship, which possess high connectivity. Interconnected gene products (nodes) were further grouped into distinct clusters based on their interaction scores (edges). Edge thickness represents the level of interaction confidence between nodes.

## Discussion

While at-arrival management practices are somewhat dependent upon anticipated risk of BRD development, both inter- and intra-herd level disease prevalence is highly variable [5, 51]. To counter this variability, beef production systems will often administer antimicrobials and/or immunostimulants at arrival to reduce the risk of clinical BRD development and associated production losses [52, 53]. However, immunostimulant administration alone as a metaphylactic protocol for controlling BRD appears to have minimal impact on rates of morbidity [54–56]. Metaphylactic use of antimicrobials at arrival reduces risk of morbidity and mortality across beef production systems, however this management practice is variable in efficacy, in both rates of disease across cattle populations and in pharmacological choice, and the practice is suspected to drive expansion of antimicrobial resistance, a growing societal concern [52, 57, 58]. Given this background, our research group and others have focused on evaluating host transcriptomes at arrival, to better characterize host-driven mechanisms and develop

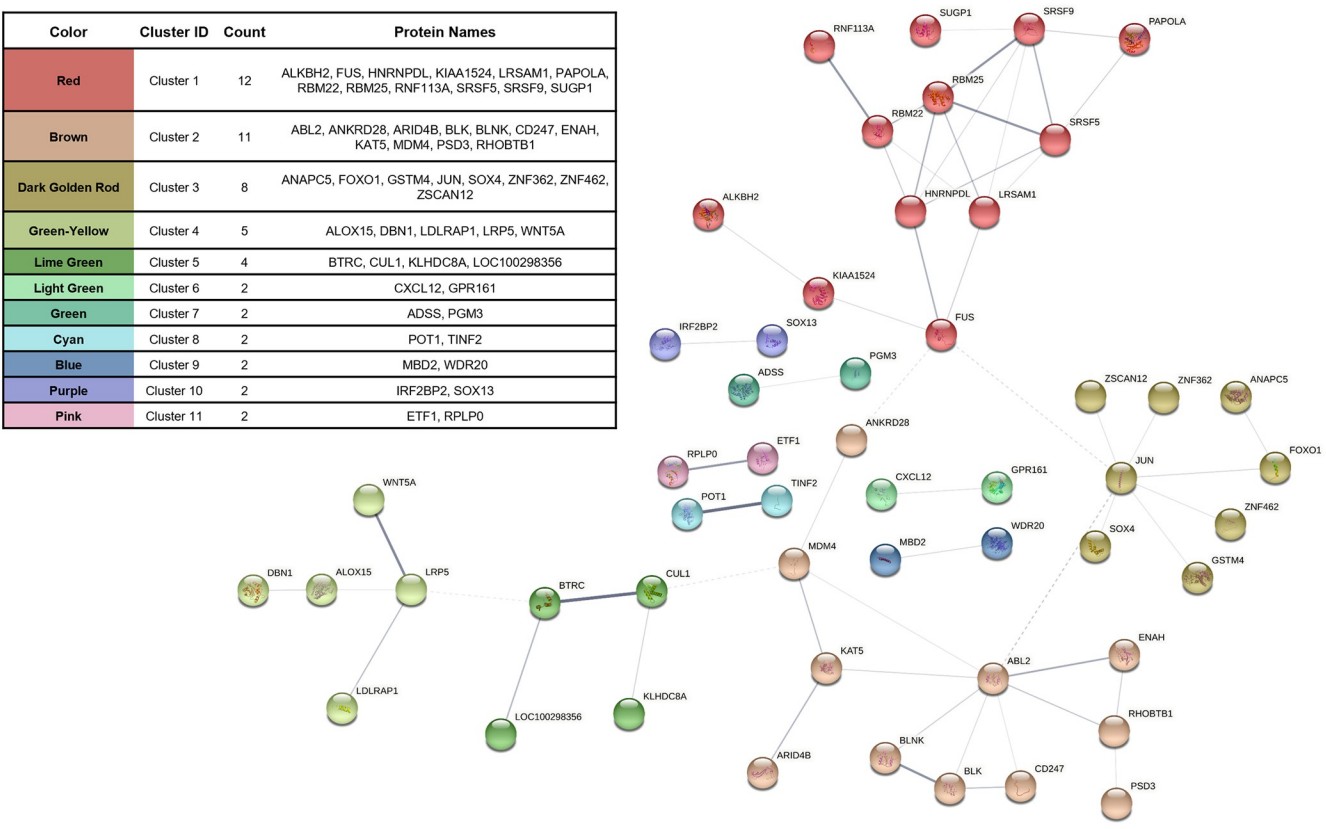

| Color | Cluster ID | Count | Protein Names |
|---|---|---|---|
| Red | Cluster 1 | 12 | ALKBH2, FUS, HNRNPDL, KIAA1524, LRSAM1, PAPOLA, RBM22, RBM25, RNF113A, SRSF5, SRSF9, SUGP1 |
| Brown | Cluster 2 | 11 | ABL2, ANKRD28, ARID4B, BLK, BLNK, CD247, ENAH, KAT5, MDM4, PSD3, RHOBTB1 |
| Dark Golden Rod | Cluster 3 | 8 | ANAPC5, FOXO1, GSTM4, JUN, SOX4, ZNF362, ZNF462, ZSCAN12 |
| Green-Yellow | Cluster 4 | 5 | ALOX15, DBN1, LDLRAP1, LRP5, WNT5A |
| Lime Green | Cluster 5 | 4 | BTRC, CUL1, KLHDC8A, LOC100298356 |
| Light Green | Cluster 6 | 2 | CXCL12, GPR161 |
| Green | Cluster 7 | 2 | ADSS, PGM3 |
| Cyan | Cluster 8 | 2 | POT1, TINF2 |
| Blue | Cluster 9 | 2 | MBD2, WDR20 |
| Purple | Cluster 10 | 2 | IRF2BP2, SOX13 |
| Pink | Cluster 11 | 2 | ETF1, RPLP0 |

**Fig 5. Protein-protein interaction network of interconnected BRD-associated hub genes.**

candidate mRNA biomarkers associated with clinical BRD outcomes [18, 19, 20]. These studies have provided valuable information regarding cattle treated based on clinical signs of BRD, but these studies heavily rely on semi-objective evaluation of BRD cases and may miss underlying subclinical or misdiagnosed disease. As such, the underlying host mechanisms involved in BRD development remain disputed. Therefore, to identify at facility arrival genes and mechanisms which may represent the variable development of BRD cases and leverage the total expression profile of individual cattle, we employed a systems biology approach with weighted co-expression network analysis. This methodology allows us to identify networks of genes exclusively co-expressed, and to evaluate said networks in a reduced manner in order to identify molecules and mechanisms of interest for future BRD prediction studies. Importantly, co-expression network analysis serves as a complementary, yet distinct, approach to identifying genes and mechanisms associated with disease status, when compared to differential expression analyses. The network approach performed in this study evaluates and identifies genes that are strongly coordinated in terms of expression, and determines correlation with overlapping metadata (clinical data), whereas differential expression analyses typically follow a pairwise approach to determine level of effect and probability of gene differences between groups. Co-expression network analyses consider greater biological context when evaluating gene expression differences, compared to more traditional pairwise approaches. Additionally, through utilization of hematological parameters, we could capture changes in the cellular composition of whole blood as they may relate to cellular and gross pathophysiology across individuals.

While we recognize that dynamic changes captured in whole blood may not completely encompass biomolecular characteristics seen within lung tissue, whole blood serves as a practical and easily obtainable sample for respiratory and inflammatory disease diagnostics [14, 59]. After initial statistical assessment of CBC data, we identified that both BRD and Healthy cattle possessed comparable lymphocytosis, thrombocytosis, and erythrocytic macrocytosis; the distribution of these values were not considered significantly different between the two groups. Notably, mild to moderate levels of dehydration, a common condition in newly arrived post-weaned beef animals, may cause elevated changes in hematocrit levels and lymphocytes [60, 61]. Lymphocytosis and thrombocytosis may also result from host responses to infection or inflammation. Additionally, reticulocytosis (i.e., immature erythrocytosis) is the most common cause of erythrocytic macrocytosis [60] and was noted as a common feature found across all blood samples submitted for analysis. While these cattle did not possess physiological nor hematological evidence of hemolysis or blood loss upon facility arrival, this finding may be associated with early regenerative anemia, systemic inflammation, or mineral deficiencies [60–62]. Furthermore, blood-borne pathogens were not reported from blood smear assessment. Nevertheless, it does not rule out the possibility of mild/subclinical intraerythrocytic pathology or asymptomatic convalescence that may result in these increased hematological changes. Such pathology is often caused by parasitic diseases such as anaplasmosis, a common infectious disease of cattle across the United States [63, 64]. It is plausible that these findings indicate that cattle at facility entry are undergoing similar physiological changes as it relates to stressful and/or pathogenic events (long-distance transportation, co-mingling, etc.) and underlying genomic mechanisms serve to resolve or prolong deleterious physiological conditions that result in BRD.

With respect to distributions, we found that the majority of variables tested for module correlations were normally distributed. Of the nine (of the 26 total) non-parametric testing variables, six demonstrated relative linearity upon visual distribution assessment (data not shown; Neu%, Lymph%, NL Ratio, MCHC, RDW, and FEC-d0). Moreover, the non-parametric nature of Eos%, Baso%, and Sex is perceived to be due to data sparsity and relative rarity of the expected cell counts attributed to eosinophils and basophils in cattle (Table 1) [65]. We elected to utilize Pearson correlation models as they can measure discrete and continuous datasets without need for transformation, and preserve linearity from the raw data structure when assessing these variables with gene co-expression modules. Additionally, calculated Pearson correlations from WGCNA can better handle datasets with missing or censored data and is highly computationally efficient [66]. We identified that RBCs were significantly increased in cattle that would go on to develop BRD versus those that did not. Although this result was identified in a relatively small number of cattle, it corresponds with the work of Richeson and colleagues [16]. As discussed within their prior research, elevated RBCs may indicate dehydration and subsequent predisposition with BRD development [5, 16]. Interestingly, we were able to identify one well-preserved co-expression module which possessed significant correlations with RBCs, RDW, PLT, BRD, and Risk_Days (i.e., shorter time to first treatment): steelblue. Upon further investigation, we discovered that the genes within this module were related to antigen receptor-mediated signaling (*BLK*, *CD247*, *CD276*, *CD3G*, *GATA3*, and *PLEKHA1*) and negative regulation of fibroblast growth factor receptor signaling (*CREB3L1*, *GATA3*, and *WNT5A*), and specifically components of alpha-beta T-cell receptor complexes (*CD247* and *CD3G*). The upregulation of IL-7R and associated signaling molecules, which include *CD3G* and *CD247*, initiate NOTCH-dependent proliferation of T-cell precursors [67]. Furthermore, elevated levels of *BLK* and *GATA3* tend to skew the immune response towards Th2-type immunity [68–70]. In terms of RBC relationship, previous research has demonstrated that Th2-stimulated bone marrow T-cells promote erythroid differentiation and lead to the

development of erythroblasts [71]. Additionally, *CXCL12*, also identified within the steelblue module and previously identified as a differentially expressed gene associated with BRD development [20], is involved in Th2-cell migration and immune response [71, 72]. *HNRNPH3*, found within the steelblue module, has previously been identified as a key transcription factor associated with clinical BRD [18]. Lastly, several genes identified in the steelblue module were also found in the "turquoise" module identified by Hasankhani and colleagues [24], which enriched positive regulation of activated T-cell proliferation and Th1/Th2-cell differentiation pathways. While this study cannot elucidate the exact mechanistic components nor temporality of molecular events, it suggests that promotion of Th2-mediated T-cells at arrival shares a common mechanism with RBC elevation and risk of BRD development. Our previous research has indicated that genes elevated at arrival in cattle that eventually develop BRD interact, and may enhance, TLR-4 and IL-6 responses [20, 40, 73], which may contribute to the co-expressed pattern related to Th2-mediated T-cell development [74]. Overall, this pattern of Th2-mediated immunity is strongly associated with clinical BRD development and timing to first treatment, and may further strengthen the depiction that early Th2 responses indicate clinical disease development and lung pathology [75, 76].

While steelblue was the only well-preserved BRD-associated module detected, four other modules were determined to be well-preserved across populations and warranted specific functional enrichment investigation: black, purple, lightgreen, and tan. Genes within the black module, largely involved with neutrophil activation and degranulation, IL-1 activity, and metabolic disease, was only significantly associated with hemoglobin and erythrocyte parameters (HGB, HCT, MCV, and MCH); notably, the black module did not possess any significant associations with clinical variables. This may indicate that neutrophilic and IL-1 activity was not indicative of BRD within this population of cattle, and/or additional disease-associated variables were not recorded in this study. Genes within the purple module, associated with increased eosinophil percentage, decreased neutrophil-lymphocyte ratio, decreased MCV and MCH, increased at-arrival fecal parasitic egg count, and increased growth rate (weight gain over 82 days), largely enriched for mitochondrial function and aerobic metabolism and RNA processing. Importantly, this module possessed positive association to weight gain independent of BRD development. Previous research has investigated many of these ribosomal protein-encoding genes for their potential for immune effector capacity [77] and cell regulation [78], however this marks the first time, to our knowledge, that they have been implicated in contributing to weight gain potential in high-risk cattle. Notably, one gene (*RPS26*) has been previously identified as a differentially decreased marker in the diseased lungs of cattle experimentally challenged with BRD-associated pathogens [79, 80]. Similar to the black module, genes identified within the lightgreen module were associated with hemoglobin and erythrocyte parameters, but additionally positively correlated with neutrophil percentage and neutrophil-lymphocyte ratio, and negatively correlated with basophil percentage; likewise, the lightgreen module did not possess significant associations with clinical variables. Lastly, the tan module, possessing several significant hematological associations, and was negatively correlated with castration status at arrival, possessed genes which enriched for B-cell receptor complexes and regulation and interferon-gamma production. Unfortunately, the underlying physiological impact of co-expressed genes identified within the black, lightgreen, and tan modules were not captured in this study. As this study was primarily focused on BRD development and severity, the genes within these three modules may possess a role in other disease complexes or immune-mediated events, such as gastrointestinal or apoptotic/necrotic diseases.

Utilizing hub gene and interaction network analyses, we further identified genes related to BRD development and severity. Here, we detected and mapped 52 genes into a protein-protein

interaction network, further stratified into 11 distinct clusters based on their combined interaction scores. This procedure helps describe the physical relationship that multiple BRD-associated gene products possess with one another in a more holistic approach. Here, we may infer that these interactions possess accompanying transient functions involved in BRD development not previously described in literature. As such, these predicted protein-protein network interactions may infer potential modular units which participate in BRD development or resistance [81, 82]. Further evidence of the associative importance related to BRD development exists with these genes, as *CXCL12* [20], *TLL2* [20], *ALOX15* [18, 20, 40], and *LOC100298356* [73, 79, 80, 83] have been previously identified as differentially expressed when comparing cattle with and without BRD development. Specifically, *CXCL12*, *TLL2*, and *LOC100298356* are considered drivers of innate surveillance and inflammation associated with BRD development, whereas *ALOX15* encodes for an enzyme involved in specialized proresolving mediator biosynthesis and associated with cattle that do not develop clinical BRD in high-risk systems [18, 20, 40, 79, 83]. Collectively, we detected these previously identified differentially expressed genes, associated to BRD development, with an independent approach. This overlap emphasizes the potential capability of these genes and mechanisms to serve a predictive role for BRD. Proteomic approaches have detailed that proteins infrequently operate as single biological entities and, when involved in similar biological functions, interact in dynamic, yet organized complexes [84–87]. As such, these findings provide candidate protein complexes related to BRD development and severity, which warrants further investigation for avenues of confirmation in larger populations of cattle and novel therapeutic target development.

## Conclusions

This study was conducted to utilize systems biology methodology to further establish genes, mechanisms, and coordinated biological complexes associated with dynamic hematological changes and BRD development. Utilizing our previously published RNA-Seq data and WGCNA, we identified five well-preserved modules of highly co-expressed genes with significant associations with hematological and clinical traits in cattle at facility arrival. The "steel-blue" module, containing genes involved in alpha-beta T cell receptor complex and negative regulation of fibroblast growth factor receptor signaling, possessed significant positive correlations with erythrocyte count, platelet count, red cell width, and BRD diagnosis, and negative correlation with days at risk for BRD. The "purple" module, containing genes involved in mitochondrial processes and non-coding RNA processing and maturation, possessed significant correlation with increased eosinophil percentage, decreased neutrophil-lymphocyte ratio, and increased growth rate (weight gain over time). Protein-protein interaction network and clustering analyses of BRD-related hub genes identified possible at-arrival biological complexes strongly associated with BRD development; many of these hub genes have been described as differentially expressed genes in previous BRD research. Through this holistic molecular approach, we provide genes, mechanisms, and predicted protein complexes associated with BRD development and performance which are warranted for future analyses targeted in predicting BRD at facility arrival.

## Supporting information

**S1 Table. Clinical metadata of cattle selected for WGCNA analysis.**
(XLSX)

**S2 Table. CBC and leukocyte distribution data of cattle selected for WGCNA analysis.**
(XLSX)

**S3 Table. Full gene list and weighted module assignment.**
(CSV)

**S4 Table. Functional enrichment analysis of the five well-preserved modules.**
(XLSX)

**S5 Table. Hub gene analysis of all five BRD-associated modules.**
(XLSX)

**S6 Table. STRING identifiers and physical interaction scores (combined_score $\geq$ 0.200).**
(XLSX)

**S1 Fig. Heatmap and hierarchical clustering of clinical and hematological data across the 23 cattle utilized for transcriptome network analysis.** Standardized connectivity was calculated from network adjacency matrices and used to classify potential outliers (Z.k < -5); no animal was identified as an outlier. The remaining rows represent the numerical values of all clinical and hematological traits across each animal. Colors indicate an increase (yellow/white) or decreased (purple/black) value for each trait; Sex and BRD are both represented as a value of 1 for bulls and Yes, and 0 for steers and No, respectively.
(TIF)

**S2 Fig. Soft threshold ($\beta$) selection for signed weighed correlation network construction through scale free topology (SFT) plot analysis.** A) SFT index $R^2$ (y-axis) at increasing soft threshold powers ($\beta$; x-axis). The value $\beta$ = 8 was selected, seen where the saturation curve is above 0.8 (orange horizontal line). B) Increasing soft threshold powers ($\beta$; x-axis) with respect to decreasing mean connectivity (y-axis). The goal of selecting a value $\beta$ is to maximize scale independence (i.e., suppress low correlations) while simultaneously minimizing loss in mean connectivity.
(TIF)

## Acknowledgments

The authors thank William Crosby, Kirsten Midkiff, Alexis Thompson, Joseph Gerlach, Merrilee Thoresen, and the supporting staff at the Mississippi State University College of Veterinary Medicine Clinical Pathology Laboratory for their technical assistance and insights throughout this study. The authors would also thank students and staff of the Mississippi Agricultural and Forestry Experiment Station (MAFES) and Mississippi State University Animal and Diary Science Department for their assistance in animal care and sample collection.

## Author Contributions

**Conceptualization:** Matthew A. Scott, Amelia R. Woolums, Andy D. Perkins, Bindu Nanduri.

**Data curation:** Matthew A. Scott, Amelia R. Woolums, Abigail Finley, Brandi B. Karisch.

**Formal analysis:** Matthew A. Scott, Abigail Finley.

**Funding acquisition:** Matthew A. Scott, Amelia R. Woolums, Cyprianna E. Swiderski, Andy D. Perkins.

**Investigation:** Matthew A. Scott, Abigail Finley, Brandi B. Karisch.

**Methodology:** Matthew A. Scott, Amelia R. Woolums, Cyprianna E. Swiderski, Andy D. Perkins, Bindu Nanduri.

**Project administration:** Amelia R. Woolums, Brandi B. Karisch.

**Resources:** Matthew A. Scott, Brandi B. Karisch.

**Supervision:** Abigail Finley, Bindu Nanduri.

**Validation:** Matthew A. Scott.

**Visualization:** Matthew A. Scott.

**Writing – original draft:** Matthew A. Scott.

**Writing – review & editing:** Matthew A. Scott, Amelia R. Woolums, Cyprianna E. Swiderski, Abigail Finley, Andy D. Perkins, Bindu Nanduri, Brandi B. Karisch.

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
