## [Decision Letter · Decision Letter 0]

13 Jul 2022

PONE-D-22-05536

Hematological and gene co-expression network analyses of high-risk beef cattle defines immunological mechanisms and biological complexes involved in bovine respiratory disease and weight gain

PLOS ONE

Dear Dr. Scott,

Thank you for submitting your manuscript to PLOS ONE. After careful consideration, we feel that it has merit but does not fully meet PLOS ONE’s publication criteria as it currently stands. Therefore, we invite you to submit a revised version of the manuscript that addresses the points raised during the review process.

We look forward to receiving your revised manuscript.

Kind regards,

Jianhua Ruan, Ph.D.

Academic Editor

PLOS ONE

https://journals.plos.org/plosone/s/file?id=ba62/PLOSOne_formatting_sample_title_authors_affiliations.pdf".

“This work was supported by the Animal Health and Disease Program [Grant no. 2020-67016-31469] from the USDA National Institute of Food and Agriculture. The authors thank William Crosby, Kirsten Midkiff, Alexis Thompson, Joseph Gerlach, Merrilee Thoresen, and the supporting staff at the Mississippi State University College of Veterinary Medicine Clinical Pathology Laboratory for their technical assistance and insights throughout this study. The authors would also thank students and staff of the Mississippi Agricultural and Forestry Experiment Station (MAFES) and Mississippi State University Animal and Diary Science Department for their assistance in animal care and sample collection.”

Reviewers' comments:

Reviewer's Responses to Questions

**Comments to the Author**

1. Is the manuscript technically sound, and do the data support the conclusions?

Reviewer #1: Partly

Reviewer #2: Yes

2. Has the statistical analysis been performed appropriately and rigorously? 

Reviewer #1: Yes

Reviewer #2: Yes

3. Have the authors made all data underlying the findings in their manuscript fully available?

Reviewer #1: Yes

Reviewer #2: Yes

4. Is the manuscript presented in an intelligible fashion and written in standard English?

Reviewer #1: Yes

Reviewer #2: Yes

5. Review Comments to the Author

Reviewer #1: 1. The manuscript cited a previous publication. However, the description of animal population was still not adequate. It would be more informative to briefly describe how animals were selected for the study in 2017 and in 2019. Was there any information about animal breed composition or major breeds? Were the animals in similar ages at arrival? If not, was the age difference significant between the BRD and Healthy animal groups? Were the cattle related by pedigree, i.e. possible family structures within the BRD or health groups? If the animal information was not available, please indicate and discuss possible impacts on the results.

2. The manuscript reports clinical data for these cattle in Supplemental Table S1. The Table has 23 cattle, which seems to be animals of 2017. How about the clinical data of cattle of 2019?

3. Supplemental Table S1 listed data of WT-d0. However, the comparison of WT-d0 was not included in Table 1. Were the average body weights at arrival significantly different between the BRD and Healthy animals? If yes, was the body weight difference associated with BRD development?

4. Line 183, please describe and define the elements in the equation.

5. Pearson correlations were calculated to assess the relationship between modules and hematological clinical traits. However, some trait values did not follow a normal distribution. Would this bias the estimation of the correlation coefficients? Please discuss.

6. The RNAseq data were analyzed previously using differential gene expression analyses, and differentially expressed genes (DEGs) related to BRD were identified. Please discuss more on how the identified DEGs in the previous study are related to results from the current study including the hub genes detected.

7. The image quality of Figure 2-5 should be improved.

Reviewer #2: The authors investigated the co-expression network analysis in bovine respiratory disease and weight gain traits.

This subject is within the scope of the journal and the results would provide insight and understanding about bovine respiratory disease and weight gain. Although the manuscript has merits in many aspects, there are several points need to be clarified, I suggest a minor revision.

1.Collectively, there are a number of grammatical typos and some spelling mistakes. Please check the manuscript carefully and revise the manuscript thoroughly.

2.Please improve the quality of the figures.

3. I need to check the WGCNA code that you used for analysis. Please put the WGCNA code in the supplementary file.

6. PLOS authors have the option to publish the peer review history of their article (what does this mean?). If published, this will include your full peer review and any attached files.

Reviewer #1: No

Reviewer #2: **Yes: **Mohammad Farhadian

---

## [Author Response · Author response to Decision Letter 0]

24 Aug 2022

Please find our document "Response to Reviewers" for all editor and reviewer responses. All additional inquiries have been addressed in the revised cover letter.

---

## [Decision Letter · Decision Letter 1]

21 Sep 2022

PONE-D-22-05536R1Hematological and gene co-expression network analyses of high-risk beef cattle defines immunological mechanisms and biological complexes involved in bovine respiratory disease and weight gainPLOS ONE

Dear Dr. Scott,

Thank you for submitting your manuscript to PLOS ONE. After careful consideration, we feel that it has merit but does not fully meet PLOS ONE’s publication criteria as it currently stands. Therefore, we invite you to submit a revised version of the manuscript that addresses the points raised during the review process.

We look forward to receiving your revised manuscript.

Kind regards,

Jianhua Ruan, Ph.D.

Academic Editor

PLOS ONE

Journal Requirements:

Reviewers' comments:

Reviewer's Responses to Questions

**Comments to the Author**

1. If the authors have adequately addressed your comments raised in a previous round of review and you feel that this manuscript is now acceptable for publication, you may indicate that here to bypass the “Comments to the Author” section, enter your conflict of interest statement in the “Confidential to Editor” section, and submit your "Accept" recommendation.

Reviewer #1: (No Response)

Reviewer #2: All comments have been addressed

2. Is the manuscript technically sound, and do the data support the conclusions?

Reviewer #1: (No Response)

Reviewer #2: Yes

3. Has the statistical analysis been performed appropriately and rigorously? 

Reviewer #1: (No Response)

Reviewer #2: Yes

4. Have the authors made all data underlying the findings in their manuscript fully available?

Reviewer #1: (No Response)

Reviewer #2: Yes

5. Is the manuscript presented in an intelligible fashion and written in standard English?

Reviewer #1: Yes

Reviewer #2: Yes

6. Review Comments to the Author

Reviewer #1: The authors have responded to the reviewer’s comments and have made some revisions. However, it is suggested that the authors incorporate the responses in the manuscript to improve clarity. For example, “ages (not recorded) were estimated to be similar upon facility arrival”, “We did not include the at arrival weight (WT-d0) as an analysis variable because there was no significant difference between the Healthy (mean: 477.0, s.d.:24.8) and BRD (mean: 475.3, s.d.:26.9) cohorts”.

Reviewer #2: I have gone through this manuscript and the response letter.

I recommended this manuscript for publication in this format.

7. PLOS authors have the option to publish the peer review history of their article (what does this mean?). If published, this will include your full peer review and any attached files.

Reviewer #1: No

Reviewer #2: **Yes: **Mohammad Farhadian

---

## [Author Response · Author response to Decision Letter 1]

3 Oct 2022

All discussion points requested by Reviewer #1 have been incorporated into the manuscript, with the addition of citations 66 and 67. References have been checked and formatted according to journal guidelines. Please see the document "Response to Reviewers" for full detail.

---

## [Editor Report · Decision Letter 2]

19 Oct 2022

Hematological and gene co-expression network analyses of high-risk beef cattle defines immunological mechanisms and biological complexes involved in bovine respiratory disease and weight gain

PONE-D-22-05536R2

Dear Dr. Scott,

We’re pleased to inform you that your manuscript has been judged scientifically suitable for publication and will be formally accepted for publication once it meets all outstanding technical requirements.

Kind regards,

Jianhua Ruan, Ph.D.

Academic Editor

PLOS ONE
---

## [Editor Report · Acceptance letter]

25 Oct 2022

PONE-D-22-05536R2 

Hematological and gene co-expression network analyses of high-risk beef cattle defines immunological mechanisms and biological complexes involved in bovine respiratory disease and weight gain 

Dear Dr. Scott:

I'm pleased to inform you that your manuscript has been deemed suitable for publication in PLOS ONE. Congratulations! Your manuscript is now with our production department. 

Kind regards, 

on behalf of

Dr. Jianhua Ruan 

Academic Editor

PLOS ONE